# Isolation of Extracellular Vesicles from Human Follicular Fluid: Size-Exclusion Chromatography versus Ultracentrifugation

**DOI:** 10.3390/biom13020278

**Published:** 2023-02-02

**Authors:** Maria Soares, Maria M. Pinto, Rui Jorge Nobre, Luís Pereira de Almeida, Maria da Graça Rasteiro, Teresa Almeida-Santos, João Ramalho-Santos, Ana Paula Sousa

**Affiliations:** 1Institute for Interdisciplinary Research (IIIUC), University of Coimbra, 3030-789 Coimbra, Portugal; 2CNC—Center for Neuroscience and Cell Biology, CIBB, University of Coimbra, Azinhaga de Santa Comba, Celas, 3004-504 Coimbra, Portugal; 3Faculty of Pharmacy, University of Coimbra, Azinhaga de Santa Comba, Celas, 3000-548 Coimbra, Portugal; 4ViraVector, University of Coimbra, 3004-504 Coimbra, Portugal; 5CIEPQPF, Department of Chemical Engineering, University of Coimbra, Pólo II, R. Silvio Lima, 3030-790 Coimbra, Portugal; 6Reproductive Medicine Unit, Centro Hospitalar e Universitário de Coimbra, Praceta, R. Prof. Mota Pinto, 3004-561 Coimbra, Portugal; 7Faculty of Medicine, University of Coimbra, Azinhaga de Santa Comba, Celas, 3000-548 Coimbra, Portugal; 8Department of Life Sciences, University of Coimbra, Calçada Martim de Freitas, 3000-456 Coimbra, Portugal

**Keywords:** follicular fluid, extracellular vesicles, ultracentrifugation, size-exclusion chromatography

## Abstract

Follicular fluid (FF) is the microenvironment where a growing oocyte develops. Intrafollicular communication ensures oocyte competence and is carried out through paracrine signaling, the exchange of molecules via gap junctions, and the trafficking of extracellular vesicles (EVs). The study of FF-derived EVs is important for both translational and fundamental research in the female reproductive field. This study aimed to compare the efficacy and purity of two EV isolation methods: size-exclusion chromatography (SEC) and ultracentrifugation (UC). EVs isolated using SEC and UC were compared regarding their size and concentration using dynamic light scattering (DLS) and nanoparticle tracking analysis (NTA); protein contamination was assessed with microBCA; specific EV markers were detected with Western blot, and EV morphology was studied with transmission electron microscopy (TEM). Our results show that although both techniques isolated small EVs, a significantly increased yield in particle number was clear with UC compared with SEC. On the other hand, SEC generated purer EVs with fewer protein contaminants and aggregates. In conclusion, the selection of the most suited approach to isolate EVs must be conducted considering the degree of recovery, purity, and downstream application of the isolated EVs.

## 1. Introduction

The ovarian antral follicle comprises a follicular fluid (FF)-filled cavity where a growing oocyte is surrounded by different somatic cells, such as *cumulus* cells, granulosa cells, and theca cells. Follicular growth is accompanied by the proliferation of somatic cells and the accumulation of FF, which consists of plasma exudate that contains extracellular vesicles (EVs), hormones, and metabolites [1]. Intrafollicular cell communication is imperative to ensure oocyte development, and FF facilitates this crosstalk that occurs through paracrine signaling, the exchange of small molecules via gap junctions, and the trafficking of EVs [2]. Studies on oocyte developmental competence [3], follicular maturation [4], and *cumulus* expansion [5] demonstrate the importance of FF-derived EVs in intrafollicular communication.

EV subtypes, including exosomes, microvesicles, and apoptotic bodies, have different sizes, morphology, and biogenesis pathways. Exosomes measure 30–150 nm and originate through the invagination of the endosomal multivesicular bodies membrane. Then, endosomal vesicles fuse with the plasma membrane, releasing the exosomes into the extracellular space [6]. Microvesicles, with dimensions ranging from approximately 100 to 1000 nm, are generated through the reorganization of the actin cytoskeleton, after which they bud directly from the plasma membrane [7]. Apoptotic bodies (50–5000 nm) form as the cell disassembles into smaller fragments during apoptosis and often contain organelle fragments [8]. 

Selecting the most suited approach to isolate EVs is challenging since the application of distinct separation methods to the same sample generates different results [9]. Indeed, human FF-derived EVs isolated using different methods resulted in different miRNA expression patterns [10,11]. The most commonly used technique for EV isolation is ultracentrifugation (UC) [12]. This method is suitable for large sample volumes but is a time-consuming and labor-intensive method that requires expensive instrumentation, which can be disadvantageous for clinical applications [13]. Moreover, it yields EV samples with low purity and integrity [14]. In contrast, size-exclusion chromatography (SEC) achieves greater purity in EV preparations. This technique utilizes chromatography columns that allow differently sized EV fractions to be eluted and is successful in isolating exosomes from biofluids [13]. SEC is less time-consuming and excludes the need to acquire expensive equipment [15]. 

UC has been used in research studies to isolate EVs from human FF [16] in the context of female aging [17] and polycystic ovary syndrome (PCOS) [18] and to predict oocyte/embryo quality for assisted reproductive treatments (ARTs) [4,10,19]. In humans, FF-derived EVs were also isolated through SEC to study miRNA expression patterns in the context of PCOS [20]. Moreover, SEC was also used to isolate EVs from bovine FF [21,22]. However, although there are many studies on FF in several species, no study has specifically compared the efficacy and purity of these two EV isolation methods from this biofluid in human samples. Therefore, this study aimed to assess the recovery and purity of EVs isolated using SEC and UC from human FF. This is of great importance to the unveiling of biomarkers and the developing of new therapeutic approaches to apply in fundamental and translational human reproductive research. 

## 2. Materials and Methods

### 2.1. Ethical Statement and Participants

The collection of ovarian FF was approved by the ethics committee of Centro Hospitalar e Universitário de Coimbra (CHUC; process number: CHUC-170-20). Written informed consent was obtained from all participants prior to FF collection at the Reproductive Medicine Unit of CHUC. The participants included in this study were 29–40-year-old women undergoing assisted reproduction techniques (ARTs) such as in vitro fertilization (IVF) or intracytoplasmic sperm injection (ICSI).

### 2.2. Ovarian Stimulation Protocol and hFF Collection 

Women received treatment to induce multiple follicle development, with either ovarian stimulation with a GnRH agonist long protocol or antagonist short protocol and using recombinant follicle-stimulating hormone (FSH) or human menopausal gonadotropin (hMG). Stimulation was monitored using the ultrasound measurement of follicle number and diameter. When at least two follicles presented a diameter greater that 17 mm, ovulation was induced with 250 µg of recombinant human chorionic gonadotrophin (rhCG). Transvaginal ultrasound-guided follicular puncture was performed 36 h after rhCG injection.

The total volume of FF from an individual participant was collected and pooled together after cumulus–oocyte complex retrieval and oocyte isolation. FF samples with massive blood contamination were excluded from further proceedings. 

### 2.3. EV Isolation from hFF

FF was centrifuged at 300 g for 15 min to remove cells, and the supernatant was stored at −80 °C until further utilization. Afterward, samples were thawed on ice and centrifuged at 2000× *g* for 20 min at 4 °C to remove cell debris and at 16,500× *g* for 45 min at 4 °C to remove large vesicles and protein aggregates. The remaining supernatant was subsequently filtered through a 0.2 µm sterile syringe filter to remove particles greater than 200 nm. From the total FF volume, 1 mL was recovered and stored at −80 °C for EV isolation using SEC. The remaining volume (26.8 ± 6.9 mL, [5.5, 35.5]) was used to isolate EVs using UC.

### 2.4. Ultracentrifugation

For EV isolation using UC, a Beckman Optima L-100 XP ultracentrifuge with a SW 28 rotor (Beckman Coulter, Brea, CA, USA) was used. The FF supernatant was ultracentrifuged in Beckman tubes (No. 326823; Beckman Coulter) at 100,000× *g* for 70 min at 4 °C to pellet EVs. Following the first ultracentrifugation, the supernatant was discarded, and freshly filtered phosphate-buffered saline (PBS; No. 18912-014; Gibco, Thermo Fisher Scientific, Waltham, MA, USA) was added to wash the pellet. The suspension was ultracentrifuged one more time at 100,000× *g* for 70 min at 4 °C to pellet the extracellular vesicles. The final pellet was resuspended in 100 µL of PBS and kept at −80 °C until further analysis. 

### 2.5. Size-Exclusion Chromatography

For EV isolation using SEC, the stored FF was thawed and centrifuged at 10,000× *g* for 10 min at room temperature (18–25 °C), and 900 µL of FF was loaded into the pre-washed qEVoriginal/70 nm chromatographic column (No. SP1; Izon Science, Christchurch, New Zealand). The SEC column was then eluted with 12 mL of freshly filtered PBS, and the eluent was recovered in 30 fractions of 500 µL using an automatic fraction collector (Izon Science). Individual fractions were concentrated using Vivaspin500 10 kDa cutoff centrifugal filter units (VS0102; Sartorius, Gottingen, Germany) at 15,000× *g* for 5–15 min at room temperature (Eppendorf 5415 D). Pooled EV-enriched SEC fractions were concentrated using Amicon Ultra-4 10 KDa cutoff centrifugal filter units (UFC801024; Merck Millipore, Burlington, MA, USA) at 3220× *g* for 50–60 min at RT. Concentrated and non-concentrated fractions were kept at −80 °C until further analysis. After the SEC protocol, all columns were washed accordingly and stored in 20% ethanol until the next utilization. Each column was reused up to five times. 

### 2.6. Dynamic Light Scattering 

The average size of the dispersed particles in solution (PBS) was estimated using photon correlation spectroscopy using Malvern Zeta Sizer Nano ZS (Marvell Panalytical, Malvern, England, UK) with a 532 nm laser. The average particle size was calculated with Zetasizer Software (Malvern; version 7.13) using a backscatter angle of 173°. UC samples and SEC fractions were equilibrated at 4 °C before starting the experiment. Each sample was measured in 1 scan, each with 3 runs.

### 2.7. Nanoparticle Tracking Analysis

Measurements of EV size and concentration were performed with NTA. A NanoSight NS300 instrument (Marvell Panalytical, Malvern) with a 488 nm laser and sCMOS camera module (Malvern Panalytical) was used following the manufacturer’s instructions. Five videos of 60 s were recorded for each sample at a camera level of 13–14. Data were processed using NTA 3.2 analytical software (Malvern) with a detection threshold of 3.

### 2.8. Total Protein Quantification Using microBCA

EV samples from UC and fractions obtained using SEC were thawed on ice and vortexed to detach EVs that might have adhered to the tube walls and to dissociate possible EV aggregates that might have formed. Protein quantification was performed using Micro BCA Protein Assay Reagent Kit (Thermo Fisher Scientific) according to the manufacturer’s instructions. Samples were measured in duplicates.

### 2.9. Western Blot

Protein samples were denatured by adding 6x concentrated SDS sample buffer (0.5 M Tris, 30% glycerol, 10.28% SDS, 0.6 M DTT, and 0.012% bromophenol blue) for 5 min at 95 °C. A total of 30 µg of protein from UC pellets and equal volumes of ~25 µL from concentrated SEC fractions were loaded per lane into 7% and 12% Acrylamide Tris-HCl gels, according to the protein sizes to be analyzed. Protein separation by means of electrophoresis was performed in a Mini protean tetra cell Bio-Rad apparatus (Bio-Rad, Hercules, CA, USA) in TGS buffer (25 mM Tris, 192 mM Glycine, and 0.1% (*w*/*v*) SDS (pH 8.3); Bio-Rad). Then, proteins were transferred onto a PDVF membrane (Immun-Blot^®^ PVDF Membranes for Protein Blotting; Cat No. 1620177; Bio-Rad) in transfer buffer (25 mM Tris, 192 mM glycine, and 20% methanol (pH 8.3); 12539S; Cell Signalling, Danvers, MA, USA) and blocked in 5% bovine serum albumin (BSA; MB04602; NZYTech, Lisbon, Portugal) Tris-Buffered Saline (137 mM NaCl and 20 mM Tris–HCl (pH 7.6)) containing 0.1% Tween20 (TBS-T) solution for 1 h at room temperature; afterwards, they were incubated overnight at 4 °C with the following primary antibodies: mouse monoclonal anti-LAMP2 (1:200; mouse monoclonal; sc-18822; Santa Cruz, CA, USA), rabbit monoclonal anti-Alix (1:500; rabbit monoclonal; ab186429; Abcam, Cambridge, England, UK), rabbit monoclonal anti-Tsg101 (1:500; rabbit monoclonal; ab125011; Abcam), anti-Calnexin (1 µg/mL; Rabbit polyclonal; ab22595; Abcam), and anti-TOM20 (1:100; mouse monoclonal; Sc-17764; Santa Cruz) diluted in 5% BSA in TBS-T. Membranes were then washed 6 times for 5 min with TBS-T and incubated for 1 h at room temperature with goat anti-mouse (1:2000; Cat No. 1706516; Bio-Rad) and anti-rabbit (1:2000; Cat No. 170-6515; Bio-Rad). Protein immunoreactive bands were visualized using Clarity Western ECL Substrate WesternBright TM Quantum (K-12042-D20; Advansta, San Jose, CA, USA) with ImagQuant LAS 500 (GE Healthcare Bio-Sciences AB, Uppsala, Sweden). Membranes were stained with Ponceau solution (0.1 % (*w*/*v*) in 5% acetic acid; P7071; Sigma, Saint Louis, MO, USA) for 5 min, rinsed in distilled water until the background was clean, and scanned with ImagQuant LAS 500 (GE Healthcare Bio-Sciences AB) for total protein detection. Ponceau staining was removed using TBS-T.

### 2.10. Transmission Electron Microscopy

EV samples obtained using UC and concentrated EV-enriched pooled fractions obtained using SEC were analyzed with a transmission electron microscope. EV samples previously fixed in 2% paraformaldehyde (Paraformaldehyde; 16% *w*/*v* aq. soln., methanol free; Thermo Fisher Scientific) were deposited onto a copper grid (300 mesh; Micro to Nano, Haarlem, Netherlands) and left to adsorb for 5 min at room temperature. Then, the grids were negatively stained with 2% uranyl acetate (Agar Scientific, Essex, England, UK) for 1 min. Observations were carried out with an FEI Tecnai G2 Spirit BioTWIN transmission electron microscope at 100 kV. 

### 2.11. Statistical Analysis 

Graphs were designed, and data were analyzed, with GraphPad Prism 9.0.0 software (San Diego, CA, USA). Diagrams are presented as mean ± SD or as a representative single experiment. In addition, when considered necessary, values are presented as mean ± SD [min, max]. Normality was evaluated with the Shapiro–Wilk test for NTA-derived values of mean, mode, and concentration. Unpaired Student’s t-test was used to compare means when normality was verified; otherwise, the Mann–Whitney test was executed to compare medians. *p* value > 0.05 equaled not significant (ns), and *p* ≤ 0.0001 ***.

## 3. Results

EVs obtained using UC and SEC fractions (Figure 1) were characterized in accordance with the Minimal Information for Studies of Extracellular Vesicles 2018 (MISEV 2018) guidelines [23]. Briefly, EV size and concentration were measured using DLS and NTA; the total protein content was analyzed with a microBCA kit; specific surface markers were studied with Western blot, and the EV morphology was examined using TEM.

### 3.1. UC EV Sample Characterization 

A series of preliminary centrifugations were performed to obtain a final EV pellet (Figure 1). Centrifugation at 300× *g*, 2000× *g*, and 16,500× *g* was conducted to pellet living cells, cellular debris, and protein aggregates, respectively. Then, by filtering the supernatant, particles larger than 200 nm were removed from the samples. Ultracentrifugation steps were conducted to pellet the EV population. Each respective supernatant was collected, and the content was studied in terms of particle size using DLS. During the differential centrifugation protocol, a 10 nm sized population (protein contamination) showed an increased percentage of intensity compared with the population of particles measuring 100 nm (Figure 2a) in most of the supernatants excepting the final EV pellet (dark blue line). This final pellet presents a single high-intensity peak at 195.5 nm ± 10.9 nm. Moreover, as depicted in Figure 2b, the mean diameter of other EV samples isolated from different subjects was 181.1 ± 26.8 nm (*n* = 8). These observations indicate that the applied differential centrifugation protocol yielded a final pellet of particles with larger diameters than what is considered for small EVs (Figure 2b). To confirm these results, EV samples from three different subjects were evaluated regarding their diameter using NTA, which showed a size mean of 165.6 ± 6.2 nm (Figure 2c).

Additionally, UC EVs presented a mean concentration of 9.01 × 10^10^ ± 1.69 × 10^10^ particles/mL. Even though the presence of small-sized particles was not detected in the final EV pellet (Figure 2c), the quantification of total protein with microBCA demonstrated that there was protein contamination in each pellet and that it was variable among samples (Figure 2d). To confirm the presence of EVs in the final pellet, samples of six different subjects were used to study specific EV markers using Western blot. The EV-associated transmembrane protein, LAMP2, and two cytosolic proteins recovered in EVs, TSG101 and ALIX (Figure 2e and Appendix A) were detected. Moreover, cellular contamination was indicated by the presence of calnexin and TOM20. Corroborating all these results, the TEM images confirmed the presence of an EV-characteristic cup-like structure in a mesh of cellular contaminants. Moreover, it was also possible to observe that the EVs were grouped in clusters instead of being independent structures (Figure 2f and Appendix A). The measurement of individual particles with TEM software showed that the isolated EVs presented diameters ranging from 80.42 nm to 185.59 nm (Appendix A).

Considering the MISEV 2018 guidelines [23], the isolated pellet contained FF-derived small EVs; however, it also included cellular contaminants, which made it a low-purity EV pellet. 

### 3.2. Total SEC Fraction Characterization

All the 30 eluted fractions were studied in terms of protein content. The first six fractions corresponded to the 3 mL of void volume and thus were not considered. Fractions 7–17 were studied with DLS to determine the particle size, and in fractions 7–12, the presence of EV markers was studied to determine which were the EV-enriched fractions. As depicted in Figure 3a, fractions 7 to 13 contained particles with diameters ranging from 50 to 150 nm, which correspond to sizes matching small EVs. It is also possible to observe that from fraction 14 on, particles presented diameters of approximately 10 nm, which indicated that protein contamination might have started from fraction 14. Moreover, one can observe that fractions 7 to 10 presented a single peak of approximately 100 nm (Figure 3b) and that from fraction 11 on, there are two differently sized populations of EVs, with one averaging 10 nm (protein contamination). 

When quantifying the total protein content in each fraction, it was possible to observe an increase in protein concentration ranging from fractions 11 to 12 (Figure 3c and Appendix A), which corroborates the previous results obtained with DLS. Additionally, the total protein content assessed using Western blot with Ponceau staining confirmed the increased presence of protein after fraction 14 (Appendix A). The presence of EV-associated markers TSG101, ALIX, and LAMP2 was found in fractions 9 to 12 (Figure 3d and Appendix A). Fractions 7 and 8 did not present any EV marker, which could be explained by an insufficient amount of EV particles contained in these fractions, as corroborated by the results of Ponceau staining (Appendix A) and protein quantification with microBCA (Figure 3c and Appendix A). Contamination with the endoplasmic reticulum protein calnexin started from fraction 9; however, the mitochondrial-related marker TOM20 was only present after fraction 11 (Figure 3d and Appendix A). Taking these results into consideration, fractions 7 to 10 were pooled together and considered EV-enriched fractions. 

#### EV-Enriched Fraction Characterization

Fractions 7 to 10 were pooled together, and an aliquot was used to calculate particle size and concentration using NTA. As shown in Figure 3e, EV-enriched fractions presented a single-peak graph, with particles having a mean size of approximately 180.2 ± 10.9 nm and a concentration of 8.55 × 10^8^ ± 3.76 × 10^8^ particles/mL. The remaining volume was then concentrated 10 × and used to study EV-associated markers and possible contaminants. Figure 3f shows the expression of LAMP2, ALIX, and TSG101, which indicated the presence of EVs in the final concentrated volume but also showed contamination with calnexin. In contrast, TOM20 contamination was not detectable (Figure 3f and Appendix A). Lesser protein contamination could also be observed using TEM, with clearer images of individual EVs (Figure 3g). 

### 3.3. EV Size and Concentration Comparison between SEC and UC

The size mean and mode, as well as the concentration of particles, were compared between the EV samples isolated using UC and EV-enriched fractions obtained using SEC. The mean diameter of EVs isolated using SEC was 180.2 ± 10.9 nm, whereas the mean diameter of EV isolated using UC was 165.6 ± 6.2 nm (Figure 4a). Regarding the average mode of EV sizes, particles isolated using SEC presented 139.9 ± 14.6 nm, and EVs obtained using UC showed 130.5 ± 10.7 (Figure 4b). Though the measured diameters of EVs isolated using UC were smaller than the size of particles obtained using SEC, there were no significant differences between the two groups. In contrast, regarding the concentration of EVs obtained using the two methodologies (Figure 4c), the number of particles isolated with UC (9.01 × 10^10^ ± 1.69 × 10^10^ particles/mL) was significantly greater than the number of particles obtained with SEC (8.55 × 10^8^ ± 3.76 × 10^8^ particles/mL). Therefore, while the methodologies showed no significant differences in terms of isolated EV sizes, UC demonstrated significantly increased recovery of particles. 

## 4. Discussion

The study of FF-derived EVs is important for both translational and basic research in the female reproduction field. Ideally, methods that combine the recovery of the greatest amount of EV particles (efficiency) without non-vesicular contaminants (purity) would improve both the relevance of the scientific findings and possible future applications. In this study, we aimed to compare the gold-standard, most used method for EV isolation, UC, with a method known to improve the purity of EV populations. Indeed, SEC has been used to improve the purity of isolated EVs in different reproductive fluids [24,25,26,27]. It was already reported that, using iZON qEV columns, it is possible to isolate an EV population with fewer non-vesicular contaminants (i.e., proteins or lipoproteins) and aggregates and increase the integrity of the vesicles [27,28,29]. Few studies have reported the use of these columns for isolating EVs from FF in women [20] and in a bovine model [21,22]. Moreover, no research has been conducted concerning the comparison of the recovery and the purity of UC and SEC in this type of biofluid. Therefore, to our knowledge, in human FF, this is the first time that a thorough characterization of EV-containing fractions was compared to EV samples isolated using the UC method.

### 4.1. FF-Derived EV Recovery

A detailed study of human FF-derived EVs demonstrated that this biofluid constitutes a very diverse environment in terms of particle morphology and size [16]. As exosomes and microvesicles show overlapping size ranges, some EV isolation methods, such as UC, frequently produce a mixture of different types of EVs, which encourages the use of nonspecific terms, such as small/large EVs [23]. However, specific EV markers that completely discriminate the two subtypes of vesicles have not yet been identified [6]. It is recognized that exosomes are enriched in tetraspanins (CD9, CD63, and CD8) and also include ESCRT components (TSG101 and ALIX) [23,30]. In contrast, microvesicles do not show unique markers [31]. 

In this study, it was shown that UC and SEC produced small EV samples and small EV-enriched fractions, respectively. To prove the presence of EV lipid bilayers, we studied the expression of LAMP2, which is a transmembrane protein associated with plasma membrane/endosomes. Moreover, to evaluate if the samples contained not only open cell fragments, proteins that are normally recovered in EVs such as ALIX and TSG101 were studied [23]. UC EVs showed the expression of those three positive markers, meaning that this technique was successful in isolating small EVs. Concerning SEC EVs, we also assessed the presence of the three markers in the F7–12 fractions eluted from the column, and we observed that LAMP2 and TSG101 markers were present from F9 and that ALIX only showed its expression from F10 onwards. F7 and F8 did not show any positive marker expression, which could be explained by the reduced content of particles and thus low protein content for detection with Western blot. As such, F7 and F8 were pooled together with F9 and F10, which clearly showed the presence of EV positive markers. When observing the final pooled EV-enriched fractions using Western blot, it was also possible to observe the presence of the three markers, meaning that SEC also produced small EV-enriched fractions. However, it presented inter-individual variability.

According to the literature, small EVs are those with diameters of 30–150 nm [6,30]. UC-isolated EVs demonstrated increased diameters in both the analyses of particle mean using DLS and NTA. This could have been due to particle aggregation, which might work as a confounding factor, increasing EV size. This hypothesis is corroborated by the TEM images, where one could observe that EVs were not independent/isolated structures, but instead formed clusters/aggregates. Indeed, the aggregation of EVs affects the isolation of vesicles and has been described as a major limitation of using UC for this purpose [27,32]. Moreover, the influence of pelleting a vesicular membrane against a solid surface at high centrifugal forces and for prolonged periods of time is still unknown in terms of how it may impact membrane integrity [33]. In contrast, the electron microscopy images of EV-enriched SEC fractions showed that EVs presented themselves as individual particles. When considering the NTA values of the EV size mode, which is the recommended measure in NTA analysis, both techniques isolated particles matching the size range of small EVs. The concentration of particles obtained in UC samples was significantly increased when compared with the number of particles in the EV-enriched fraction obtained using SEC. This result is expectable, since the starting volume of UC was greater than the volume loaded into the column. Additionally, it was already reported that even though SEC purifies EVs, it also dilutes them, which corroborates our findings [29,34]. Moreover, as UC showed greater contamination with non-vesicular particles such as proteins or lipoproteins, it is likely that the NTA overestimated the amount of EVs, confusing them with contaminating particles.

To sum up, the UC samples and EV-enriched SEC fractions presented typical EV markers (LAMP, ALIX, and TSG) demonstrating that the lipid bilayer was intact and that the particles contained the typical proteins of EVs; the TEM images showed the characteristic cup-like structures of small EVs in both UC samples and SEC fractions; and the mode size matched the diameters of small EVs (50–150 nm) when assessed with NTA. Taking these results into consideration, both techniques successfully isolated small EVs from human FF, with UC demonstrating greater EV recovery due to the possibility of using greater initial volumes. 

### 4.2. FF-Derived EV Purity

To study the purity of the UC samples and SEC fractions, we quantified their total protein concentration with microBCA. All pellets had some protein contamination; however, there was high inter-individual variability in protein content. This could be attributed to different initial volumes of the samples as well as other physiological differences among subjects. Concerning the SEC fractions, there was a slight increase from F7, which was indicative of some non EV-associated protein contamination. Protein contamination increased from fraction 12, with the possible EV-enriched fractions corresponding to F7 to F12.

To complement the purity studies, the measurement of the particle sizes was conducted in both UC samples and SEC fractions using DLS. All UC samples showed a single peak at around 100 nm, meaning that protein contamination was not detectable by DLS equipment. In contrast, when analyzing the individual SEC fractions, a two-sized peak starting from F11 was observed, with fractions F7 to F10 being those that only presented particles with diameters matching the small EV size range. 

To further study the purity of the UC samples and SEC fractions, we evaluated the presence of calnexin, a reticulum endoplasmic-related protein, and TOM20, a mitochondrial-related protein, which are considered EV negative markers and are expected to be absent in EV populations [23]. Considering the UC samples, all subjects presented cellular contamination given by the expression of calnexin and TOM20. Thus, UC alone does not yield pure EVs, particularly when they are isolated from complex biofluids, as is the case of FF. Small EVs are pelleted at high centrifugal forces (i.e., 100,000–120,000× *g*), but such UC step cannot totally separate EVs by size, since sedimentation also depends on the particle density and the distance the particles travel towards the bottom of the centrifugal tube. For instance, small contaminants might pellet along the small EVs, and even with resuspending and recentrifuging steps, absolute separation is difficult to achieve with UC alone [14]. 

Moreover, we also studied the “exclusion markers” in SEC fractions F7–12 and observed that calnexin contamination started from F9 and TOM contamination started from F11. Therefore, we chose to pool F7 and F8, which did not show any cellular contamination, with F9 and F10, which only showed contamination with the endoplasmic reticulum-related protein. F7–12 were then considered the EV-enriched fractions with less cellular contamination. This was corroborated using Western blot analysis, which showed no contamination with TOM20 and partial contamination with calnexin in the pooled fractions. 

One way to improve the purity of EV samples by removing lipoproteins and protein contaminants could be the sequential utilization of two or more isolation techniques. However, this methodology is correlated with a significant decrease in EV yield [35]. 

According to MISEV 2018 [23], the isolation methodology must be chosen according to the downstream application of EVs. Taking into consideration the aforementioned results, it is possible to conclude that even though SEC presents a decreased yield of EVs compared with UC, it provides a purer population of particles in the final EV-enriched fractions. This observation corroborates already published work where authors separate EVs according to their size ranges to acquire pure and functional EVs [34,36]. 

## 5. Conclusions

This study compared the efficacy and purity of two EV isolation methods from human FF: SEC and UC. For the first time, EV recovery from human FF using SEC was extensively characterized regarding particle size, concentration, surface markers, morphology, and protein contaminants. UC yields significantly more EV particles than the SEC method; however, the final pellet appears to be more contaminated with non-vesicular cellular components. For future information, both techniques successfully isolate EVs from hFF; however, the choice of the technique depends on the available initial volume of sample, the degree of purity, and the concentration of particles that are required for the intended purpose of the work.

## Figures and Tables

**Figure 1 biomolecules-13-00278-f001:**
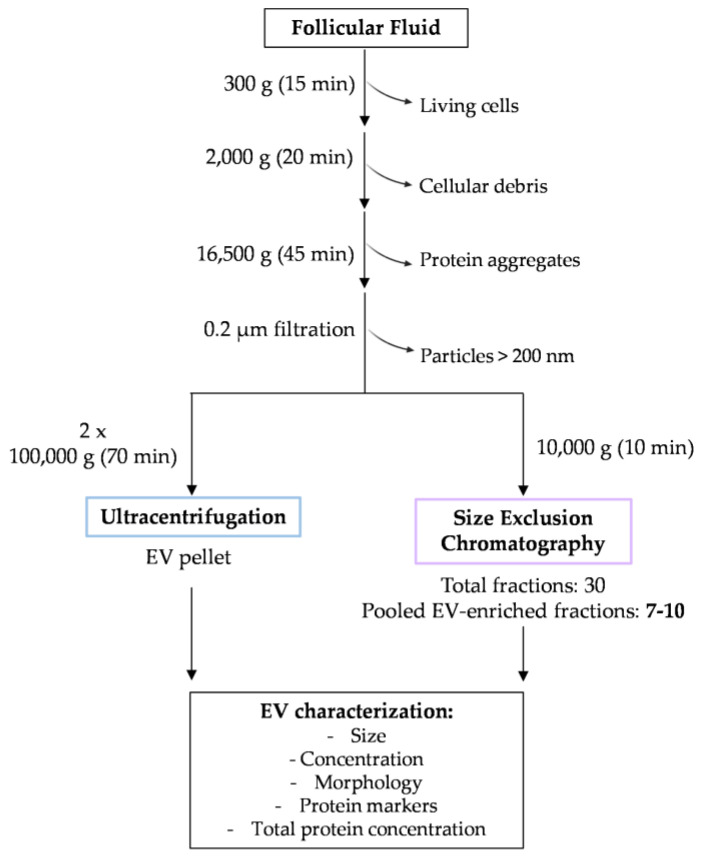
Schematic representation of the experimental design. FF samples were centrifuged at 300× *g* and 2000× *g* to exclude living cells and cellular debris, respectively. The supernatant was centrifuged at 16,500× *g* to pellet protein aggregates. A 0.2 µm filtration step was conducted to remove particles larger than 200 nm. From this volume, 1 mL was stored at −80 °C for EV isolation using SEC, and the remaining volume was ultracentrifuged twice to isolate the final UC EV pellet. In total, 30 fractions of 500 µL were collected, and EV-enriched fractions (F7–10) were used for further characterization. The EVs isolated using UC and SEC F7–10 fractions were characterized regarding the particle size, concentration, morphology, presence of specific markers, and the total protein content using DLS, NTA, TEM, Western blot, and microBCA, respectively.

**Figure 2 biomolecules-13-00278-f002:**
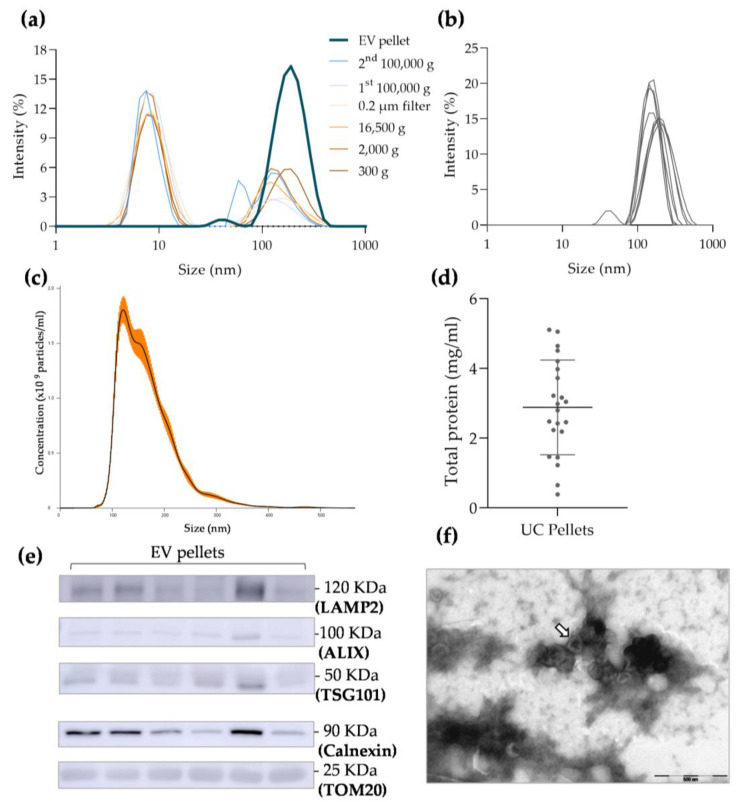
Characterization of extracellular vesicle (EV) samples isolated using ultracentrifugation (UC): (**a**) dynamic light scattering (DLS) analysis of particle size distribution in each supernatant of differential centrifugation protocol and EV final pellet; (**b**) DLS analysis of particle size distribution in follicular fluid (FF)-derived UC pellets (*n* = 9); (**c**) nanoparticle tracking analysis (NTA) of particle concentration and size distribution in FF-derived UC pellets (*n* = 3; orange bars indicate ± SEM); (**d**) total protein quantification of each EV pellet using microBCA kit (*n* = 22); (**e**) Western blot membrane showing the expression of specific EV positive markers, such as LAMP2, ALIX, and TSG101, and also the presence of cellular contaminants in the final UC pellet, such as Calnexin and TOM20 (*n* = 6; see Appendix A for controls); (**f**) representative image of EVs isolated using UC obtained with transmission electron microscopy observations (EV cup-like structure indicated by the arrow).

**Figure 3 biomolecules-13-00278-f003:**
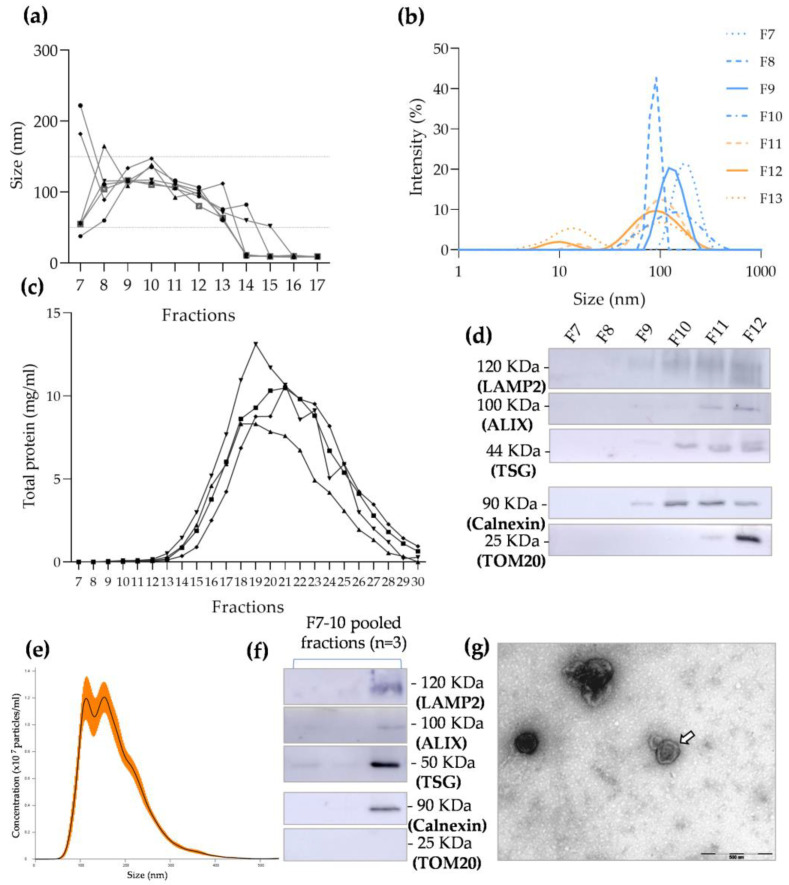
Characterization of total fractions and extracellular vesicle (EV)-enriched fractions obtained using size-exclusion chromatography (SEC): (**a**) dynamic light scattering (DLS) analysis of particle sizes in fractions 7 to 17, where the ranging size of small EVs (50–150 nm) is highlighted by the dotted line and values correspond to the highest-intensity population in each fraction; (**b**) DLS analysis of particle size distribution in fractions 7 to 13, where the orange lines represent fractions containing two differently sized peaks of intensity and the blue lines correspond to a one-intensity-peak EV population (*n* = 1); (**c**) graph demonstrating total protein quantification in fractions 7 to 30 (*n* = 4); (**d**) Western blot membrane showing the expression of specific EV positive markers, such as LAMP2 (fractions 9 to 12), ALIX (fractions 10 to 12), and TSG101 (fractions 9 to 12), and the presence of calnexin and TOM20 contamination in fractions 9 to 12 and fractions 11 to 12, respectively (see Appendix A for controls); (**e**) nanoparticle tracking analysis (NTA) of particle concentration and size distribution in EV-enriched fractions (F7–10), where the orange bars indicate ± SEM; (**f**) Western blot membrane showing the expression of specific EV positive markers (LAMP2, ALIX, and TSG101) and calnexin contamination in EV-enriched fractions, but no detectable presence of TOM20 (see Appendix A for controls); (**g**) representative transmission electron microscopy image of the EV-enriched fractions isolated using SEC (EV cup-like structure indicated by the arrow).

**Figure 4 biomolecules-13-00278-f004:**
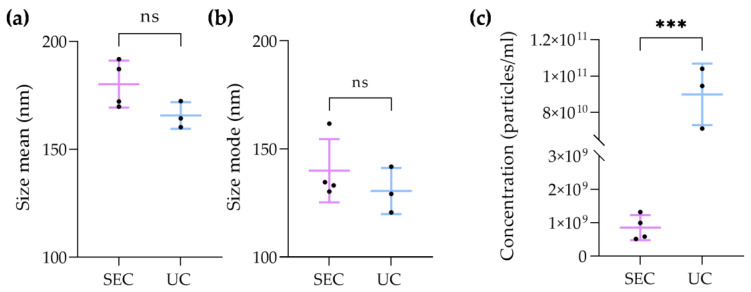
Comparison of EV diameter size and mode, and particle concentration between ultracentrifugation (UC) and size-exclusion chromatography (SEC): (**a**) graph showing the comparison diameter mean of EVs obtained with UC (*n* = 3) and SEC (*n* = 4) (unpaired *t*-test); (**b**) graph demonstrating the comparison of EV diameter mode of EVs isolated with UC (*n* = 3) and SEC (*n* = 4) (Mann–Whitney test); (**c**) graph presenting the difference between the concentration of particles isolated with UC (*n* = 3) and SEC (*n* = 4) (unpaired *t*-test). *p* value > 0.05 equals not significant (ns), and *p* ≤ 0.0001 ***.

## Data Availability

Not applicable.

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
