# Peer review of "Isolation of Extracellular Vesicles from Human Follicular Fluid: Size-Exclusion Chromatography versus Ultracentrifugation"

_biomolecules, 2023, doi:10.3390/biom13020278_

Round 1

Reviewer 1 Report

This paper is exclusively a technical article highlighting the different potentials of two analytical methods (SEC and UC). The biological role and functions of EVs from FF are not discussed by the Authors. The article results in a mere characterization of vesicles, not even so complete in terms of a multi-omics approach.

Over the past 10 to 15 years, much has been written and learned about EV isolation techniques. There are so many articles on the advantages/disadvantages of using one technique over another.

Line 455-456: “For the first time, EV recovery from human FF through SEC was extensively characterized regarding particle size, concentration, surface markers, morphology, and protein contaminants.” It is not correct to say "for the first time" there are several other published studies about the characterization of Follicular fluid. For example: Int. J. Mol. Sci. 2022, 23, 11676. https://doi.org/10.3390/ijms231911676 and Int. J. Mol. Sci. 2020, 21, 5365; doi:10.3390/ijms21155365.

Therefore, the article is not applicable for Biomolecules 

Author Response

Reply: We understand the Reviewers point of view, it is true that there have been many studies on this biological fluid, in human as in other species (such as bovine). We in no way, disregard this previous work and, indeed, we cited both papers the Reviewer specifically mentions in their report. Over the past years, several authors aimed to unveil the best methodology to isolate EVs from follicular fluid. In humans, there are studies using UC to isolate FF-derived EVs (Martinez 2018, doi:10.1038/s41598-018-35379-3; Neyroud 2022, doi:10.3390/ijms231911676) and others that try to extract EVs with Izon SEC columns (Rooda 2020; doi:10.3390/ijms21249550). Similarly, those methodologies were tested in FF from animal models (Hasan 2020, 10.3390/ijms21155365). However, none of the studies compared the efficacy and purity of UC and SEC isolation in human follicular fluid. We believe this has merit and that our findings (more volume of EVs using one method, higher purity in another) may be important in eventually establishing translational protocols using FF. To better highlight this idea, the last paragraph of the introduction was altered accordingly.

Reviewer 2 Report

This is an interesting study which directly compared two EV isolation methods: UC vs. SEC, in human follicular fluid (FF). One strength of the work is that the comparison was made by using patient derived FF. The work is straightforward and the results are within expectation. I have the following comments:

(1) Figure 1, under "EV characterization", list bullet points of what you characterized in this work.

(2) Line 246, "considering minimal information...". Not clear what authors want to present here. 

(3) Figure 2a, the colors of different lines are confusing. Please improve. 

(4) Figure 2c, what is the "orange area"?

(5) Figure 2e, lack internal control

(6) Figure 2f, please label/indicate "cup-like" EV

(7) Figure 3b, color of lines are confusing.

(8) Figure 3d & f, lack internal control

(9) Figure 3e, what is the orange area?

(10) Figure 3g, label/indicate "cup-like" EV structure

(11) Figure 4, what additional information does the "mode" panel provide (in addition to the "mean")?

Author Response

This is an interesting study which directly compared two EV isolation methods: UC vs. SEC, in human follicular fluid (FF). One strength of the work is that the comparison was made by using patient derived FF. The work is straightforward, and the results are within expectation.

Reply: Thank you for your thoughtful comments, they are addressed below.

I have the following comments:

(1) Figure 1, under "EV characterization", list bullet points of what you characterized in this work.

Reply: We considered the suggestion, and the figure was altered accordingly.

(2) Line 246, "considering minimal information...". Not clear what authors want to present here.

Reply: The text was altered to clarify our intention to cite the Minimal Information for Studies of Extracellular Vesicles (MISEV) guidelines.

(3) Figure 2a, the colors of different lines are confusing. Please improve.

Reply: According to the suggestion, we improved the figure by widening the y-axis and highlighting the line representing the EV pellet. This graph shows the single-peaked EV pellet line that contrasts to the other supernatants that show two peaks.

(4) Figure 2c, what is the "orange area"?

Reply: The orange area corresponds to the error bars that indicate ± 1 standard error of the mean. This information was added to the legend of figure 2c.

(5) Figure 2e, lack internal control

Reply: For the sake of space and to keep the figures to a readable size the controls were included in the supplementary materials. We hope this is acceptable

(6) Figure 2f, please label/indicate "cup-like" EV

Reply: The figure was altered according to the reviewer suggestion. We added an arrow in the figure and the respective text in the legend.

(7) Figure 3b, color of lines are confusing.

Reply: We altered the figure to highlight the difference between the double-peaked lines in orange and the single-peaked lines in blue.

(8) Figure 3d & f, lack internal control

Reply: We apologize, again the controls were included in the supplementary materials.

(9) Figure 3e, what is the orange area?

Reply: The orange area corresponds to the error bars that indicate ± 1 standard error of the mean. This information was added to the legend of figure 3e.

(10) Figure 3g, label/indicate "cup-like" EV structure

Reply: The figure was altered according to the reviewer suggestion. We added an arrow in the figure and the related text in the legend.

(11) Figure 4, what additional information does the "mode" panel provide (in addition to the "mean")?

Reply: The NanoSight NTA manual state that typically the modal particle size is used to describe the sample. We added the mean sizes to provide additional information about the particles.

In fact, the mean size may consider confounding measurements such as very small protein contaminates or large EVs aggregates. Therefore, the mode size better describes the sample as it provides us information regarding the most frequent particle population of a given size.

Author Response

After reading the manuscript “Isolation of Extracellular Vesicles from Human Follicular Fluid: Size Exclusion Chromatography versus Ultracentrifugation”, I have some following comments for authors:

  1. Line 232, authors mentioned about a mean concentration of 9.01x1010 particles/mL (Figure 2c). But I can’t find these numbers in this figure. Moreover, in Figure 2c, what is the meaning of numbers on the vertical axis? What are they expressed for ( millions, billions…)? Authors should add a number on the top of the vertical axis, for example: x1010 to explain the magnitude of the numbers.

Reply: We thank the Reviewer for their insightful comments. The figure was altered accordingly.

  1. Line 272: “from fraction 10” compared to Figure 3b, it should be “from fraction 11”.
  2. Line 278-279: please add a name of Figure that explains the sentence “The presence of …TSG101, ALIX, and LAMP2 was found in fractions 9 to 12.”

Reply: The text was altered accordingly.

  1. Line 292: authors mentioned about a concentration of 8.55x108 particles/mL (Figure 3e). But I can’t find these numbers in this figure. Please add a number on the top of the vertical axis in this Figure as I mentioned in the No.1.

Reply: We considered the suggestion, and the figure was altered accordingly.

Round 2

Reviewer 1 Report

Specify that it is a TECHNICAL REPORT

Reviewer 2 Report

Thank you for the revision.